# Did the COVID-19 Pandemic Truly Adversely Affect Disease Progress and Therapeutic Options in Breast Cancer Patients? A Single-Centre Analysis

**DOI:** 10.3390/jcm11041014

**Published:** 2022-02-15

**Authors:** Tomasz Nowikiewicz, Maria Szymankiewicz, Marta Drzewiecka, Iwona Głowacka-Mrotek, Magdalena Tarkowska, Magdalena Nowikiewicz, Wojciech Zegarski

**Affiliations:** 1Department of Surgical Oncology, Nicolaus Copernicus University in Toruń, Collegium Medicum in Bydgoszcz, 85-067 Bydgoszcz, Poland; zegarskiw@cm.umk.pl; 2Department of Clinical Breast Cancer and Reconstructive Surgery, Oncology Centre-Prof. Franciszek Łukaszczyk Memorial Hospital, 85-796 Bydgoszcz, Poland; m.drzew@wp.pl; 3Department of Microbiology, Oncology Centre-Prof. Franciszek Łukaszczyk Memorial Hospital, 85-796 Bydgoszcz, Poland; szymankiewiczm@co.bydgoszcz.pl; 4Department of Rehabilitation, Nicolaus Copernicus University in Toruń, Collegium Medicum in Bydgoszcz, 85-094 Bydgoszcz, Poland; iwona.glowacka@cm.umk.pl; 5Department of Urology, Nicolaus Copernicus University in Toruń, Collegium Medicum in Bydgoszcz, 85-094 Bydgoszcz, Poland; magdalena.sowa@cm.umk.pl; 6Department of Hepatobiliary and General Surgery, A. Jurasz University Hospital, 85-094 Bydgoszcz, Poland; magdalena.nowikiewicz@gmail.com

**Keywords:** breast cancer, diagnostic and treatment, COVID-19 pandemic, cohort study

## Abstract

Purpose: The uncontrolled spread and transmission of SARS-CoV-2 infections has disrupted most areas of social and economic life all over the world. The most important changes concern problems related to the functioning of healthcare systems. The aim of this study was to evaluate clinical consequences associated with the COVID-19 pandemic for patients with newly diagnosed breast cancer, treated at our centre. Methods: The study participants were patients first time diagnosed with breast cancer, treated between January 2019 and March 2021, who were provided any type of cancer treatment at our centre. The study determined the grade of clinical and pathological progress of the disease and types of cancer treatment applied in patients. Results: In total, 2863 patients were included in the analysis. The number of hospitalized patients was 1228 (1123 treated surgically, 105 receiving conservative treatment) in 2019, 1318 (1206 and 112 patients, respectively) in 2020, and 317 (288 and 29 patients, respectively) in 2021. Conclusions: Despite many hazards associated with the new epidemiological situation, we were able to maintain the continuous operation of our centre. We have achieved a measurable success, and even managed to increase the number of treated breast cancer patients.

## 1. Introduction

On 11 March 2021, one year passed since the World Health Organization announced the global COVID-19 pandemic [1]. The spread of COVID-19 disrupted daily life for people across the globe. Healthcare systems quickly became overloaded and unable to meet the exponentially increasing demand of infected patients [2]. Many report that cancer referrals and clinical pathways have been substantially disrupted during the COVID-19 pandemic.

Delays during oncological treatment have been classified as primary, secondary, and tertiary. A significant proportion of time is usually consumed in primary (interval between symptom onset to the first visit to the clinician) and secondary delay (interval between clinician visit to start of treatment), in a majority of oncology patients awaiting treatment.

Since the time factor is of paramount importance in oncology, delivering optimal care during the pandemic has been a challenging event [3,4,5].

A number of steps were undertaken in response to this new situation. For example, scientific associations modified their recommendations on the management of cancer patients [6,7], and experts have taken to monitoring the secondary impacts of the COVID-19 pandemic by measuring the influence on this group of patients [8,9].

The aim of this study was to evaluate clinical consequences associated with the pandemic for patients with newly diagnosed breast cancer, treated at our centre.

## 2. Material and Methods

### 2.1. Patient Group

The study comprised newly diagnosed breast cancer patients (diagnoses from groups C50 and D05 according to ICD-10), treated between January 2019 and March 2021, who were provided any type of cancer treatment at our centre. The study was approved by the Ethics Committee at the Nicholas Copernicus University in Toruń, Collegium Medicum in Bydgoszcz (KB 312/2021 of 18 May 2021). Due to the retrospective character of the analysis, patient written consents to participation in the study were not required under the decision of the Ethics Committee.

Taking into account the moment, in which restrictions related to the COVID-19 pandemic were introduced, the analysed period was divided into two stages: January 2019–March 2020—before the pandemic (group I); and April 2020–March 2021—after its announcement (group II).

Treatment plans for patients were established in accordance with the generally applicable standards of therapeutic management for breast cancer [10,11].

### 2.2. Evaluated Clinical Data

Clinical data required for the analysis came from a prospectively maintained database and from patients’ medical hospital records.

The study determined: (1) the grade of clinical and pathological progress of the disease [12]; (2) types of cancer treatment applied in patients.

Statistical analysis was conducted using IBM SPSS Statistics 27 data analysis software. Between-groups differences were assessed using a Student’ *t*-test for continuous variables and a chi-square test (H0 Wald statistics) for categorical variables. Data are presented as a mean ± standard deviation, or a number of cases and percentage where appropriate. In all statistical analyses, the cut-off value for probability coefficient was set at *p* value ≤ 0.05.

## 3. Results

A total of 2863 patients were included in the analysis. The number of hospitalized patients was 1228 (1123 treated surgically, 105 receiving conservative treatment) in 2019, 1318 (1206 treated surgically, and 112 receiving conservative treatment) in 2020, and 317 (288 and 29 patients, respectively) between January and March 2021. Detailed clinical data of patients referred to surgical treatment is shown in Table 1.

In the second period, the number of patients treated surgically remained on a similar level. Thus, our investigation suggests no important influence of the pandemic on distribution of patients by type of surgical treatment (Figure 1).

Out of the total of patients, 9.1% (135/1490) and 8.1% (111/1373) were referred to non-surgical treatment in the first and second period of the study, respectively. Surgical treatment was mainly impossible due to spread of the disease. De novo stage IV breast cancer was found in 58.5% (79/135) and 69.4% (77/111) patients from group I and group II, respectively. This concerned 57.1% (60/105), 62.5% (70/112) and 89.7% (26/29) of patients provided with non-surgical treatment in 2019, 2020, and 2021, respectively. In the remaining cases, patients either did not consent to surgical treatment, were of poor performance status, had progressed on the disease during systemic treatment, or had a coexisting advanced non-operative cancer condition at initial presentation.

Although the rate of newly diagnosed cases prior to the pandemic was consistent with actual trends, characteristics of the disease were more advanced.

## 4. Discussion

A significant increasing trend in incidence of newly diagnosed breast cancer was detected in Poland [13]. It is to be noted that our hospital is proud to treat the largest number of newly diagnosed cancer patients, and this phenomenon was reflected in our centre. Kujawsko-Pomorskie Voivodeship, in which the Centre of Oncology in Bydgoszcz operates, is a typical industrial and agricultural region of the country, with a number of inhabitants (2.07 million), a population density (116 people/km^2^), and a percentage of urban areas (61.1%) at an average level.

Contrary to data indicating a decreasing number of cancer patients referred to oncology centres [3,4,5], the situation found in ours was quite different. When compared to the situation before the pandemic, 15.6% more patients referred to surgical treatment were hospitalized at the second stage of the study (when comparing 12-month periods April 2019–March 2020, and April 2020–March 2021: 1092 and 1262 patients, respectively). At the same time, the number of patients receiving the conservative treatment did not change (110 and 111 patients, respectively, in the same 12-month periods). However, in that case, the number of patients receiving the palliative treatment increased.

Different data was presented by Gathani et al. [14], who found that the number of patients beginning their treatment for breast cancer within the first six months of 2020 was lower by 16% versus the corresponding period in 2019. A similar decreasing trend was observed by Baxter et al., who described a reduction of nearly 20% in the number of patients receiving systemic treatment, observed during the first 4 months of the pandemic [4]. Other authors provided similar findings [4,15,16,17]. Did we manage to achieve something noteworthy?

We believe that, despite the ongoing challenging situation, we maintained our modus operandi. We introduced a number of restrictions (i.e., forbidding hospital visits, patients were asked not to leave their rooms, and our hospital was the only place of work for the personnel at the time) and additional requirements (i.e., all patients were tested for SARS-CoV-2 infection prior to hospitalization, the food distribution system was adapted and boxed meals were delivered to patients replacing dining in the hospital cafeteria, hospitalizations were shortened when no longer needed, and there was a reintroduction of the early detection of breast cancer by mammography programme). We are certain that similar actions were implemented in other healthcare centres. However, in our case they proved to be highly effective.

The most important consequence of the pandemic was a higher clinical staging of newly diagnosed neoplasms. The differences (highly statistically significant: stage I—*p* = 0.003; stage II—*p* = 0.001) result from a progress in the primary tumour size observed during the pandemic (in a clinical: *p* = 0.033 and pathological: *p* = 0.048 evaluation). At the same time, this did not follow for the most important prognostic factor in breast cancer patients—axillary lymph node status [18] (no significant difference in that respect was found in both compared groups).

Referral to the surgery was not responsible for an increased percentage of mastectomies (by 5.5%—*p* = 0.004) during the pandemic. It did not result from a decreased use of axillary radiotherapy (associated with the breast-conserving therapy) by the patients or by the therapeutic team. It was, however, closely related to the above-mentioned change in the primary tumour size.

Unfortunately, no changes were implemented on a national level. Therefore, the relative merit of increased number of breast cancer cases treated in our centre during the pandemic should be noted. We introduced strict protocols for the management of patients with breast cancer, their evaluation and treatment.

We believe that it is of paramount importance to strengthen national evidence-informed guideline programs and offer the quality of our evidence to support healthcare recommendations. Patients should not be left with the concern of obtaining access to treatment.

## 5. Conclusions

The ongoing COVID-19 pandemic has changed the healthcare industry profoundly and created unexpected delays in oncologic treatment. Due to the improvement in pandemic situation (i.e., widespread access to vaccinations, very-high level of coverage in testing), we can now focus on rebuilding the nations’ healthcare workforce. The negative impact of the COVID-19 pandemic on oncology and the volume of missed cancer-related service is yet to be fully assessed [19].

## Figures and Tables

**Figure 1 jcm-11-01014-f001:**
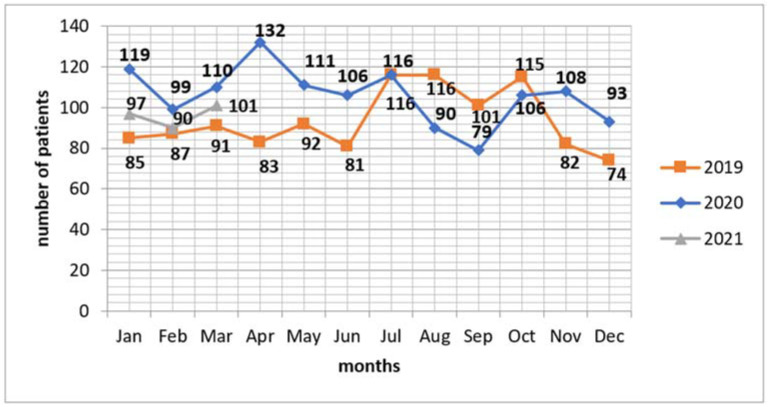
Patients referred to surgical treatment.

**Table 1 jcm-11-01014-t001:** Clinical characteristics of patients referred to surgical treatment.

Clinical Data Analysed	Group I (Pre-Pandemic Period)	Group II (Pandemic Period)	*p*
	*n* = 1355	*n* = 1262	
Age (range)	59.2 ± 12.2 (27–91)	58.8 ± 12.3 (24–90)	0.392
Histological form of cancer:			0.369
-Invasive	1261 (93.1)	1189 (94.2)	
-Preinvasive (DCIS, LCIS)	94 (6.9)	73 (5.8)	
Palpability of a tumour	891 (65.9)	924 (73.3)	<0.001
Tumour size—clinical evaluation [mm] (range)	26.0 ± 18.9 (5–200)	27.3 ± 19.0 (4–150)	0.033
Tumour size—clinical evaluation:			
-cT0	2 (0.2)	0	0.172
-cT1	662 (48.9)	528 (41.8)	<0.001
-cT2	528 (39.0)	565 (44.8)	0.003
-cT3	78 (5.8)	100 (7.9)	0.028
-cT4	82 (6.1)	66 (5.2)	0.363
Metastatic lesions—clinical evaluation:			
-cN0	1119 (82.6)	1043 (82.6)	0.966
-cN1	204 (15.1)	194 (15.4)	0.821
-cN2	24 (1.8)	21 (1.7)	0.833
-cN3	8 (0.6)	4 (0.3)	0.301
Clinical stage (cTNM):			
-I (I A)	621 (45.8)	505 (40.0)	0.003
-II:	610 (45.0)	647 (51.3)	0.001
II A	453 (33.4)	469 (37.2)	0.046
II B	157 (11.6)	178 (14.1)	0.054
-III:	116 (8.6)	103 (8.2)	0.713
III A	34 (2.5)	41 (3.3)	0.257
III B	75 (5.5)	59 (4.7)	0.319
III C	7 (0.5)	3 (0.2)	0.248
-IV	8 (0.6)	7 (0.6)	0.904
Tumour size—pathological evaluation:	*n* = 1006	*n* = 911	
-pT0	1 (0.1)	0	0.341
-pTis	94 (9.3)	73 (8.0)	0.302
-pT1	510 (50.7)	460 (50.5)	0.930
pT1mic	16 (1.6)	27 (3.0)	0.043
pT1a	53 (5.3)	36 (4.0)	0.171
pT1b	149 (14.8)	113 (12.4)	0.126
pT1c	292 (29.0)	284 (31.2)	0.306
-pT2	370 (36.8)	341 (37.4)	0.768
-pT3	13 (1.3)	25 (2.7)	0.023
-pT4	18 (1.8)	12 (1.3)	0.406
Tumour size—pathological evaluation [mm] (range)	20.2 ± 19.0 (1–260)	21.1 ± 14.4 (1–110)	0.048
Metastatic lesions—pathological evaluation:			
-pN0	651 (64.7)	622 (68.3)	0.099
-pN1mi	24 (2.4)	19 (2.1)	0.658
-pN1a	161 (16.0)	145 (15.9)	0.958
-pN2a	60 (6.0)	38 (4.2)	0.075
-pN3a	37 (3.7)	28 (3.1)	0.465
-pNx	73 (7.3)	59 (6.5)	0.501
Neoadjuvant treatment	349 (25.8)	351 (27.8)	0.235
BCT	890 (65.7)	760 (60.2)	0.004
Mastectomy	465 (34.3)	502 (39.8)	
ACT	913 (67.4)	850 (67.4)	0.988
ALND	442 (32.6)	412 (32.6)	
Tumour diagnosed during screening mammography *	334/763 (43.8)	287/711 (40.4)	0.185
Patient place of residence (voivodeship):			0.626
-Kujawsko-Pomorskie	1138 (84.0)	1051 (83.3)	
-Other	217 (16.0)	211 (16.7)	

* In Poland screening mammography is eligible for women aged between 50 and 69 years.

## Data Availability

Patient level data are available upon request from the separate participating cancer registry. Access to the full dataset will be provided if each participating cancer registry gives permission for data sharing. Statistical code is available upon request with the corresponding author. Consent from participants was not obtained but the presented data are anonymised and risk of identification is low.

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
