# Peer review of "Did the COVID-19 Pandemic Truly Adversely Affect Disease Progress and Therapeutic Options in Breast Cancer Patients? A Single-Centre Analysis"

_jcm, 2022, doi:10.3390/jcm11041014_

Round 1

Reviewer 1 Report

The authors present a single institution description of cases and breast cancer presentations over the peri-pandemic time period. The manuscript is largely descriptive and not particularly novel, but may be of interest to public health authorities in the region.

Author Response

Dear Sir or Madam,

thank you for your kind opinion. I marked revised parts of the manuscript in red. I hope you will find these satisfactory. 

Kind regards
Tomasz Nowikiewicz

Reviewer 2 Report

This is a retrospective, observational, monocentric study aimed to compare the number of patients with newly diagnosed breast cancer treated at the authors’ Institution before and during the COVID-19 pandemic (Jan 2019-March 2020 vs. April 2020-March 2021, respectively).

Overall, 2863 patients were included in the present analysis (2617 with early breast cancer, 246 with de novo metastatic breast cancers). The number of hospitalized patients did not significantly change between the two groups (before and during the pandemic), thus the authors conclude that they were “able to maintain the continuous operation of our center. We have achieved a measurable success, and even managed to increase the number of treated breast cancer patients.”

I want to thank the authors for the effort of effectively managing cancer patients in this difficult period due to COVID-19 pandemic, as well as for the effort of collecting and sharing these data.

Overall, the manuscript is well written, and the level of English language is acceptable.

My major comments would address the results section. This section is very short, and solely based on the comparison of the number of hospitalized patients between the two groups (before and during the pandemic). While one may agree with the authors’ interpretation of these numbers, showing no striking differences in the total number of hospitalized patients in the two periods, I think that some considerations could be done observing the specific subgroups. As reported in Table 1, some statistically significant differences can be observed between the two groups, e.g., palpability of a tumor (p<.001), and tumor size (both clinical evaluation, p=.033, and pathological evaluation, p=.048). These differences seem to indicate that tumors diagnosed during the pandemic tended to be larger, and this might reflect a delayed diagnosis. Consistently, the authors report that “Distant metastases were diagnosed in 58.5% (79/135) and 69.4% (77/111) patients from group I and group II, respectively”, showing a higher rate of advanced disease in the pandemic group. Therefore I would suggest the authors to expand the results section and further dissect the differences between the two groups, without limiting to the simplistic comparison of the total number of the two groups.

Please find here below some additional minor comments:

  • Page 4, lines 90-92: the authors say “Out of the total number of patients, 9.1% (135/1490) and 8.1% (111/1373) were qualified for the conservative treatment at the first and the second stage of the study, respectively. Surgical treatment was mainly impossible because of a diagnosis of cancer spread”. I would suggest to re-phrase this sentence to make it clearer to the reader. Consider to replace “conservative treatment” with “non-surgical treatment”, or “systemic treatment”, in order to avoid misunderstandings with conservative surgery. I would also suggest to indicate the precise number of de novo metastatic patients (instead of “Surgical treatment was mainly impossible because of a diagnosis of cancer spread”).
  • Page 4, lines 92-94: the authors say “Distant metastases were diagnosed in 58.5% (79/135) and 69.4% (77/111) patients from group I and group II, respectively.” Are you referring to metastases found at diagnosis (de novo metastatic patients) or later during the disease course? Please clarify.
  • Page 4, lines 94-95: “This concerned 57.1% (60/105), 62.5% (70/112) and 94 89.7% (26/29) of patients provided conservative treatment in 2019, 2020, and 2021, respectively. In the remaining cases, conservative treatment was a consequence of not giving consent to surgical treatment or a generally poor condition of patients, breast cancer progress during the inductive systemic treatment or second cancer in a different site.” Please consider to replace “conservative treatment” with “non-surgical treatment”, or “systemic treatment”, in order to avoid misunderstandings with conservative surgery.
  • Page 4, line 100: “Although the presented data comes from only one centre of oncology, yet it may be treated as representative for the whole Poland”. I would be cautious in such comparisons, if not supported by evidence-based data. I would suggest to present this consideration in the form of a hypothesis, or just to remove it and to focus on the discussion of the presented data.
  • Page 4, line 120: “So, did we manage to achieve something impossible in our case?” same comment as above.
  • Table 1: “tumor diagnosed during screening mammography”: these numbers are reported as a ratio in both groups (334/763 and 287/711): could you better clarify what the numbers refer to? Number of tumors on total number of mammographies (in healthy women)? Or number of tumors on number of patients who underwent a mammography in both groups?
  • Please consider to add (also as supplementary material) a table or a figure showing the numbers of screening mammographies performed across the years (before and during the pandemic).

Author Response

Dear Sir or Madam,

thank you for your intricate opinion. I addressed all your remarks and marked revised parts of the manuscript in red. I hope you will find these satisfactory. 

  1. Results section has been expanded.

  2. Page 4, lines 90-92: the authors say “Out of the total number of patients, 9.1% (135/1490) and 8.1% (111/1373) were qualified for the conservative treatment at the first and the second stage of the study, respectively. Surgical treatment was mainly impossible because of a diagnosis of cancer spread”. I would suggest to re-phrase this sentence to make it clearer to the reader. Consider to replace “conservative treatment” with “non-surgical treatment”, or “systemic treatment”, in order to avoid misunderstandings with conservative surgery. I would also suggest to indicate the precise number of de novo metastatic patients (instead of “Surgical treatment was mainly impossible because of a diagnosis of cancer spread”).

    Response:
    Out of the total of patients, 9.1% (135/1490) and 8.1% (111/1373) were referred to non-surgical treatment in the first and second period of the study, respectively. Surgical treatment was mainly impossible due to spread of the disease. De novo stage IV breast cancer was found in 58.5% (79/135) and 69.4% (77/111) patients from group I and group II, respectively. This concerned 57.1% (60/105), 62.5% (70/112) and 89.7% (26/29) of patients provided non-surgical treatment in 2019, 2020, and 2021, respectively. In the remaining cases, patients either did not consent to surgical treatment, were of poor performance status, had progressed on the disease during systemic treatment, or had coexisting advanced non-operative cancer condition at initial presentation.

  3. Page 4, lines 92-94: the authors say “Distant metastases were diagnosed in 58.5% (79/135) and 69.4% (77/111) patients from group I and group II, respectively.” Are you referring to metastases found at diagnosis (de novo metastatic patients) or later during the disease course? Please clarify.

    Response:
    De novo stage IV breast cancer was found in 58.5% (79/135) and 69.4% (77/111) patients from group I and group II, respectively. 

  4. Page 4, lines 94-95: “This concerned 57.1% (60/105), 62.5% (70/112) and 94 89.7% (26/29) of patients provided conservative treatment in 2019, 2020, and 2021, respectively. In the remaining cases, conservative treatment was a consequence of not giving consent to surgical treatment or a generally poor condition of patients, breast cancer progress during the inductive systemic treatment or second cancer in a different site.” Please consider to replace “conservative treatment” with “non-surgical treatment”, or “systemic treatment”, in order to avoid misunderstandings with conservative surgery.

    Response:
    This concerned 57.1% (60/105), 62.5% (70/112) and 89.7% (26/29) of patients provided non-surgical treatment in 2019, 2020, and 2021, respectively. In the remaining cases, patients either did not consent to surgical treatment, were of poor performance status, had progressed on the disease during systemic treatment, or had coexisting advanced non-operative cancer condition at initial presentation. 

  5. Page 4, line 100: “Although the presented data comes from only one centre of oncology, yet it may be treated as representative for the whole Poland”. I would be cautious in such comparisons, if not supported by evidence-based data. I would suggest to present this consideration in the form of a hypothesis, or just to remove it and to focus on the discussion of the presented data.

    Response:
    A significant increasing trend in incidence of newly diagnosed breast cancer was detected in Poland [13]. It is to be noted that our hospital is proud to treat the largest number of newly diagnosed cancer patients and this phenomenon was reflected in our centre. 

  6. Page 4, line 120: “So, did we manage to achieve something impossible in our case?” same comment as above.

    Response:
    A significant increasing trend in incidence of newly diagnosed breast cancer was detected in Poland [13]. It is to be noted that our hospital is proud to treat the largest number of newly diagnosed cancer patients and this phenomenon was reflected in our centre. 

  7. Table 1: “tumor diagnosed during screening mammography”: these numbers are reported as a ratio in both groups (334/763 and 287/711): could you better clarify what the numbers refer to? Number of tumors on total number of mammographies (in healthy women)? Or number of tumors on number of patients who underwent a mammography in both groups?

    Response:
    In Poland screening mammography is eligible for women aged between 50 and 69 years.

  8. Please consider to add (also as supplementary material) a table or a figure showing the numbers of screening mammographies performed across the years (before and during the pandemic).

    Response:
    Data on screening mammography has been collected, with statistical analysis pending.

Kind regards
Tomasz Nowikiewicz

Reviewer 3 Report

The current study evaluated clinical consequences associated with the COVID-19 pandemic for patients with newly diagnosed breast cancer in Kujawsko-Pomorskie Voivodeship, Centre of Oncology, Bydgoszcz, Poland. 15.6% more cancer patients qualified for surgical treatment were hospitalized during the pandemic compared to the pre-covid 12-months timeframe, while other regions like the UK and Scotland reported 16% to 20% reductions in cancer patients qualified for surgical treatments. This was possible due to the  strict implementations of additional requirements to prevent the spread of covid in the hospital settings. 

Figure 1: Y-axis, correct the spelling, x-axis convert the roman numerals to Jan, Feb, Mar etc in coherence with the study where months rather than numerals have been used.

The authors should consider studying the impact of covid-19 on the cohort of patients with cancer. This will provide understanding from the patient's point of view and the lessons that we may carry forward. For e.g., https://www.mp.pl/paim/issue/article/15925/ and  https://www.frontiersin.org/articles/10.3389/fpsyg.2021.647196/full

Author Response

Dear Sir or Madam,

thank you for your intricate opinion. I marked revised parts of the manuscript in red. I hope you will find these satisfactory. 

  1. Figure 1: Y-axis, correct the spelling, x-axis convert the roman numerals to Jan, Feb, Mar etc in coherence with the study where months rather than numerals have been used.

    Response:
    Figure 1 - corrected

  2. The authors should consider studying the impact of covid-19 on the cohort of patients with cancer. This will provide understanding from the patient's point of view and the lessons that we may carry forward. For e.g., https://www.mp.pl/paim/issue/article/15925/ and  https://www.frontiersin.org/articles/10.3389/fpsyg.2021.647196/full

    Response:
    Our study focused on integrated cancer strategies implemented in our center during the COVID-19 pandemic. Gaining the patient perspective on COVID and how best to respond to it is of different issue and therefore has not been analysed.

Kind regards
Tomasz Nowikiewicz

Round 2

Reviewer 2 Report

Thank you for addressing my comments and implementing the relative changes in the manuscript. To further improve the work, I would suggest an editing of English language.